# Pre-Existing Diabetes and COVID-Associated Hyperglycaemia in Patients with COVID-19 Pneumonia

**DOI:** 10.3390/biology10080754

**Published:** 2021-08-05

**Authors:** Andrea Laurenzi, Amelia Caretto, Chiara Molinari, Elena Bazzigaluppi, Cristina Brigatti, Ilaria Marzinotto, Alessia Mercalli, Raffaella Melzi, Rita Nano, Cristina Tresoldi, Giovanni Landoni, Fabio Ciceri, Vito Lampasona, Marina Scavini, Lorenzo Piemonti

**Affiliations:** 1San Raffaele Diabetes Research Institute, IRCCS Ospedale San Raffaele, Via Olgettina 60, 20132 Milan, Italy; laurenzi.andrea@hsr.it (A.L.); caretto.amelia@hsr.it (A.C.); molinari.chiara@hsr.it (C.M.); bazzigaluppi.elena@hsr.it (E.B.); brigatti.cristina@hsr.it (C.B.); marzinotto.ilaria@hsr.it (I.M.); mercalli.alessia@hsr.it (A.M.); melzi.raffaella@hsr.it (R.M.); nano.rita@hsr.it (R.N.); lampasona.vito@hsr.it (V.L.); scavini.marina@hsr.it (M.S.); 2Molecular Hematology Unit, IRCCS Ospedale San Raffaele, Via Olgettina 60, 20132 Milan, Italy; tresoldi.cristina@hsr.it; 3Department of Anesthesia and Intensive Care, IRCCS Ospedale San Raffaele, Via Olgettina 60, 20132 Milan, Italy; landoni.giovanni@hsr.it; 4School of Medicine, Università Vita-Salute San Raffaele, Via Olgettina 58, 20132 Milan, Italy; ciceri.fabio@hsr.it; 5Hematology and Bone Marrow Transplantation Unit, IRCCS Ospedale San Raffaele, Via Olgettina 60, 20132 Milan, Italy

**Keywords:** COVID-19, diabetes, clinical outcome, humoral response, SARS-CoV-2

## Abstract

**Simple Summary:**

COVID-associated hyperglycaemia is emerging as a complication of Sars-CoV-2 infection, and this clinical entity still needs to be adequately characterized in comparison to pre-existing diabetes. Few studies have comparatively characterized these two conditions in relation to the presence of comorbidities, pre-hospitalization treatments, symptoms at admission, and laboratory variables associated with COVID-19 severity. Our study generated several interesting findings. Patients with COVID-associated hyperglycaemia had significantly less comorbidities, increased levels of inflammatory markers, and indicators of multi-organ injury than those with pre-existing diabetes, while islet autoimmunity prevalence and anti-SARS-CoV-2 antibody responses were similar. COVID-associated hyperglycaemia was associated with a poorer clinical outcome and a longer viral clearance time compared to pre-existing diabetes. This strongly supports the need to screen all COVID-19 patients for hyperglycaemia at the time of admission despite a mute personal or family history of diabetes and to treat them in order to reach and maintain good glycemic control during hospitalization for COVID-19 pneumonia.

**Abstract:**

Aim. The aim of the current study was to compare clinical characteristics, laboratory findings, and major outcomes of patients hospitalized for COVID-19 pneumonia with COVID-associated hyperglycaemia or pre-existing diabetes. Methods. A cohort of 176 adult patients with a diagnosis of pre-existing diabetes (*n* = 112) or COVID-associated hyperglycaemia (*n* = 55) was studied. Results. Patients with COVID-associated hyperglycaemia had lower BMI, significantly less comorbidities, and higher levels of inflammatory markers and indicators of multi-organ injury than those with pre-existing diabetes. No differences between pre-existing diabetes and COVID-associated hyperglycaemia were evident for symptoms at admission, the humoral response against SARS-CoV-2, or autoantibodies to glutamic acid decarboxylase or interferon alpha-4. COVID-associated hyperglycaemia was independently associated with the risk of adverse clinical outcome, which was defined as ICU admission or death (HR 2.11, 95% CI 1.34–3.31; *p* = 0.001), even after adjustment for age, sex, and other selected variables associated with COVID-19 severity. Furthermore, at the same time, we documented a negative association (HR 0.661, 95% CI 0.43–1.02; *p* = 0.063) between COVID-associated hyperglycaemia to swab negativization. Conclusions. Recognizing hyperglycaemia as a specific clinical entity associated with COVID-19 pneumonia is relevant for early and appropriate patient management and close monitoring for the progression of disease severity.

## 1. Introduction

“New-onset” hyperglycaemia and acute metabolic decompensation of pre-existing diabetes are commonly recognized phenomena associated with past coronavirus infections [1,2]. During the Middle East Respiratory Syndrome Coronavirus (MERS-CoV) outbreak in 2012 and the Severe Acute Respiratory Syndrome Coronavirus-1 (SARS-COV-1) outbreak in 2003, hyperglycaemia was a commonly observed finding, even in patients without a prior history of diabetes and who did not use glucocorticoids and was an independent predictor for mortality [1,2]. Not unexpectedly, pre-existing diabetes also resulted in an association with poor outcomes and death after SARS-CoV-2 infection [3,4,5], and “new-onset” hyperglycaemia [6] and acute metabolic decompensation of pre-existing diabetes [7] are not uncommonly recognized phenomena associated with COVID-19 pneumonia, especially in patients requiring hospitalization. Moreover, isolated cases of true new-onset diabetes have been reported during COVID-19 [8,9,10,11,12,13,14]. These finding suggest the existence of a bidirectional link between COVID-19 and diabetes [15]. Both reduced insulin secretion and increased insulin resistance have been proposed as possible mechanisms [16,17]. SARS-CoV-2 might exert direct cytotoxicity against beta cells with a diabetogenic effect [18,19]. Whether beta cells express ACE2 and TMPRSS2 (two factors associated with SARS-CoV-2 infection [20]) and are permissive to SARS-CoV-2 infection is debated [21,22,23,24,25,26]. However, there is a common agreement that SARS-CoV-2 can be detected in the pancreas where it may cause inflammation that secondarily affects beta cells, and pancreatic enlargement, abnormal amylase or lipase levels, and pancreatitis were described in critically ill COVID-19 patients [27,28,29]. On the other hand, the surge in cytokine production associated with viral infection may be responsible for an increase in insulin resistance [30]. Regardless of the mechanism responsible for the metabolic dysregulation observed, the clinical entity of COVID-associated hyperglycaemia still has not been adequately characterized and separated from the entity of pre-existing diabetes [6]. Few studies have comparatively characterized the two conditions for the presence of comorbidities, pre-hospitalization treatments, symptoms at admission, and laboratory variables associated with the severity of infection. Many studies have recently reported that COVID-associated hyperglycaemia is associated with a poorer outcome compared to that of the normoglycaemic individuals [31,32,33,34,35,36]. However, whether COVID-associated hyperglycaemia is associated with a poorer clinical outcome compared to pre-existing diabetes is still an open question, and there have been conflicting findings [32,34,37,38,39]. Moreover, data on islet autoimmunity prevalence, anti-SARS-CoV-2 antibody responses, and timing for viral clearance are still missing. Finally, it is unclear whether COVID-associated hyperglycaemia persists or reverts when the viral infection resolves. To address this gap in our knowledge, we studied a cohort of 176 adult patients with confirmed COVID-19 pneumonia who also had a diagnosis of pre-existing diabetes or COVID-associated hyperglycaemia.

## 2. Material and Methods

### 2.1. Study Population and Data Sources

The study population consisted of 176 adult patients hospitalized for COVID-19 pneumonia with a diagnosis of pre-existing diabetes or hyperglycaemia. These patients were selected from two previously described and partially overlapping cohorts [31] and were assembled at the IRCCS San Raffaele Hospital (Milan, Italy) as part of an institutional clinical–biological cohort (COVID-BioB; ClinicalTrials.gov Identifier: NCT04318366) of patients with COVID-19 pneumonia. The first cohort (*n* = 584) was previously analysed for their humoral response against SARS-Cov-2 (*n* = 509), while the second cohort (*n* = 169) was studied for the development of thromboembolic complications. The study was approved by the local Institutional Review Board (Comitato Etico—Ospedale San Raffaele; protocol n 34/int/2020; NCT04318366), and all patients signed a written informed consent. A confirmed case was defined as the presence of symptoms and radiological findings suggestive of COVID-19 pneumonia and/or a SARS-CoV-2-positive RT-PCR test from a nasal/throat (NT) swab. Demographic information, clinical features, and laboratory tests were obtained within 72 h of admission. Data were collected directly through patient interviews or medical chart reviews. Data regarding patient comorbidities were collected during hospitalization or after discharge from both structured baseline patient interviews and hospital paper or electronic medical records. All disease diagnoses present at the date of infection were included as comorbidities. Data were verified by data managers and clinicians for accuracy and were crosschecked in blind. Data were recorded until hospital discharge or in-hospital death, whichever occurred first. We also recorded mortality and swab negativization beyond hospital discharge through our dedicated follow-up clinic: for patients who did not attend the follow-up clinic, we verified the patient’s vital status with their either family members or their family physician.

### 2.2. Laboratory Variables

Routine blood tests encompassed serum biochemistry (including renal and liver function, lactate dehydrogenase (LDH)), complete blood count with differential, markers of myocardial damage (troponin T and pro-brain natriuretic peptide (proBNP)), and inflammation markers (C-reactive protein (CRP), ferritin, erythrocyte sedimentation rate (ESR), interleukin-6 (IL-6), procalcitonin). An extended coagulation profile was obtained, which included D-dimer, PT, PTT, fibrinogen, and antithrombin activity. Specific antibodies to different SARS-CoV-2 antigens, interferon alpha-4 and glutamic acid decarboxylase (GAD) were tested by a luciferase immunoprecipitation system (LIPS) assay, as previously described [31].

### 2.3. Definition of Diabetes

Study participants were defined as having (a) pre-existing diabetes if they had a documented diagnosis of diabetes before hospital admission for COVID-19 pneumonia (fasting plasma glucose (FPG) ≥ 7.0 mmol/L or HbA1c ≥ 6.5% (48 mmol/mol), or if they were prescribed diabetes medications); or (b) COVID-associated hyperglycaemia if they had a mean fasting plasma glucose in the absence of infusions of dextrose ≥7.0 mmol/L during hospitalization for COVID-19 pneumonia in the presence of a negative history for diabetes and/or normal glycated haemoglobin level in the last year and in the absence of prescribed diabetes medications. We computed mean fasting glucose and glucose variability (standard deviation) from all fasting laboratory glucose values measured during hospitalization. The diagnosis of pancreatogenic and steroid-induced diabetes was attributed to patients whose diabetes was reasonably secondary to their exocrine pancreatic disease or steroid use. The diagnosis of type 1 diabetes was attributed to patients whose diabetes was associated at onset with evidence of islet autoimmunity. The diagnosis of type 2 diabetes was attributed to all patients who did not fulfil the criteria for any of the above.

### 2.4. Statistical Analysis

Categorical variables are reported as frequency or percent, and continuous variables are reported as median with the inter-quartiles range (IQR) in parenthesis. Categorical variables were compared using the Chi-square test or Fischer’s exact test as appropriate; continuous variables were calculated using the Mann–Whitney test. Associations between baseline variables and diabetes status was assessed by logistic regression. Multiple regression models, each adjusted for age and sex and one additional baseline variable of interest, were performed. The effect estimates were reported as Odd Ratios (ORs), and the outcome variables were pre-existing diabetes or new onset diabetes to obtain OR > 1, as appropriate.

Survival was estimated according to the Kaplan–Meier test. The time-to-event was calculated from the date of symptom onset to the date of the event or of the last follow-up visit, whichever occurred first. We used univariate and multivariate Cox proportional hazards models to study the association between patient characteristics with time to adverse outcome (a composite endpoint of admission to ICU or death, whichever occurred first). The effect estimates were reported as Hazard Ratios (HRs) with the corresponding 95% CI estimated according to the Wald approximation. Multivariate analyses were performed including variables significant at the level of <0.1 in the univariate analysis. Variables were excluded from multivariate Cox regression if they showed substantial biological redundancy with other variables (e.g., aspartate aminotransferase vs. alanine aminotransferase) or had data obtained for fewer than 78% of patients in order to prevent model overfitting. The two-tailed P values are reported, with a *p* value < 0.05 indicating statistical significance. All confidence intervals are two-sided and were not adjusted for multiple testing. Statistical analyses were performed with the SPSS 24 (SPSS Inc./IBM) and R software version 3.4.0 (R Core Team (2020)).

## 3. Results

### 3.1. Study Participants

The study population consisted of 176 adult patients (≥18 years) with pre-existing diabetes or COVID-associated hyperglycaemia admitted between 25 February and 2 May 2020 to the Emergency or Clinical departments of the IRCCS San Raffaele Hospital (Milan, Italy). Among the 176 cases included in the study, 11 (6.2%) were discharged without hospitalization, while 126 (71.6%) were treated with non-invasive ventilation, and 39 (22.2%) were admitted to the ICU over the hospitalization period. The median hospital stay was 17 (8–17) days. As of 14th February 2021, the median follow-up time after symptom onset was 213 (95% CI: 199–227) days. A total of sixty-four patients died (36.4%). According to the composite endpoint of ICU admission or death, 82 patients (46.6%) had an adverse in-hospital outcome.

### 3.2. COVID-Associated Hyperglycaemia Was Associated with Specific Basal Characteristics

Pre-existing diabetes and COVID-associated hyperglycaemia accounted for 68.8% (*n* = 121) and 31.25% (*n* = 55) of our cohort, respectively. Among the 121 subjects with pre-existing diabetes, 109 (90.1%) were diagnosed with type 2 diabetes, 4 (3.3%) were diagnosed with with type 1 diabetes, and 8 (6.6%) were diagnosed with secondary diabetes (5 pancreatogenic, 3 steroid-induced). The characteristics of the study participants for patients with pre-existing diabetes or COVID-associated hyperglycaemia at the time of hospitalization are reported in Table 1. HbA1c levels were available for some patients at admission and were significantly higher in pre-existing diabetes patients than in COVID-associated hyperglycaemia patients, confirming the recent onset of glucose alteration in COVID-associated hyperglycaemia. Sex- and age-adjusted logistic regression analysis was used to assess the associations between baseline variables and pre-existing diabetes vs. COVID-associated hyperglycaemia. Higher BMI at admission [after log1p transformation OR 19.44 (2.204–171.43), *p* = 0.008] cardiovascular comorbidities [OR 3.16 (1.32–8.01), *p* = 0.017], hypertension [OR 2.14 (1.04–4.4), *p* = 0.038], active neoplastic disease [OR 3.69 (1.045–1.305), *p* = 0.043], and chronic kidney disease [OR 2.23 (0.897–5.56), *p* = 0.084] were all associated with pre-existing diabetes, while comorbid neurodegenerative diseases were associated with COVID-associated hyperglycaemia [OR 3.23 (1.044–10.02), *p* = 0.042]. The median time from symptom onset to hospital admission was 6 (2–9.5) and 7 (3–10) days for patients with pre-existing diabetes and COVID-associated hyperglycaemia, respectively (*p* = 0.484). Patients with pre-existing diabetes and COVID-associated hyperglycaemia reported similar symptoms at the time of hospital admission for COVID-19 pneumonia (Table 1).

### 3.3. COVID-Associated Hyperglycaemia Was Associated with a Specific Clinical Laboratory Profile at Admission

Upon admission (Table 2), patients with COVID-associated hyperglycaemia exhibited a significantly higher white blood cell count (9.7 (6.1–14.3) vs. 7.1 (5.4–10.4) × 10^9^/L, *p* = 0.004), neutrophil count (8.4 (4.85–12.9) vs. 5.2 (3.9–8.1) × 10^9^/L, *p* = 0.003), and tissue damage markers (LDH 8.04 (5.35–16.63) vs. 5.92 (4.42–8.17) µkat/L, *p* = 0.001; AST 0.95 (0.65–1.39) vs. 0.63 (0.42–1.14) µkat/L, *p* = 0.001; ALT 0.78 (0.47–1.24) vs. 0.53 (0.3–0.93) µkat/L, *p* = 0.012) compared to patients with pre-existing diabetes. Furthermore, an increase in procalcitonin (0.95 (0.49–3.58) vs. 0.61 (0.31–1.38) ng/mL, *p* = 0.014) and hemoglobin (12.9 (11.5–14.6) vs. 12.5 (10.75–13.5) g/L, *p* = 0.042) was evident, with a trends also being present for ferritin (1254 (585–2433) vs. 823 (462–1499) µg/L, *p* = 0.063) and total bilirubin (11.62 (8.16–18.81) vs. 9.23 (5.39–14.87) µmol/L, *p* = 0.054). All the other laboratory variables associated with COVID-19 pneumonia severity, including inflammation indices (CRP and IL-6), hemostatic variables (D-dimer levels, platelet count, fibrinogen, partial thromboplastin time, and prothrombin time), biomarkers of myocardial damage (troponin T and Pro-BNP), markers of liver and kidney function (albumin, phosphatase alkaline, creatinine), and glucose control were similar between pre-existing diabetes and COVID-associated hyperglycaemia. Antibody responses to SARS-CoV-2 were available for a large subgroup of patients (Table 2). No differences between pre-existing diabetes and COVID-associated hyperglycaemia were evident for the IgG, IgM, and IgA responses to the SARS-CoV-2 spike protein (RBD or S1 + S2) or in IgG response to NP, autoimmune antibodies anti GAD, and anti-interferon alpha-4.

### 3.4. COVID-Associated Hyperglycaemia Was Associated with a Worse Clinical Outcome

Sex-and age-adjusted Cox proportional hazards model (HR 2.11, CI 1.34–3.31; *p* = 0.001) and the Kaplan–Meier estimator log-rank test (*p* = 0.005) indicated that COVID-associated hyperglycaemia was strongly associated with an increased risk of adverse clinical outcome, as defined by composite endpoint of admission to ICU or death, whichever occurred first. (Figure 1 and Table 2). Laboratory variables associated with adverse clinical outcome in a univariate Cox proportional hazards model included inflammation indices (CRP, IL-6, ferritin, procalcitonin, low lymphocytes count, high white blood cell, and neutrophil count), haemostatic parameters (D-dimer levels, fibrinogen, and partial thromboplastin time), biomarkers of myocardial damage (troponin T and Pro-BNP), markers of liver and kidney dysfunction (albumin, total bilirubin and creatinine), markers of tissue damage (lactate dehydrogenase and aspartate transaminase), and markers of glucose control (fasting plasma glucose and glucose variability). Of note, the presence of hypertension (HR 0.59, CI 0.36–0.94; *p* = 0.028) and treatment with Angiotensin II receptor blocker (HR 0.44, CI 0.22–0.85; *p* = 0.014) was associated with a decreased risk of adverse clinical outcome. A multivariable analysis using two different models confirmed COVID-associated hyperglycaemia (HR 2.68, CI 1.47–4.89; *p* = 0.001 and HR 1.72, CI 0.95–3.09; *p* = 0.073, respectively) as an independent predictor of adverse clinical outcome (Figure 2) for patients with COVID-19 pneumonia. To evaluate the specific impact of clinical characteristics and laboratory tests, the univariate Cox proportional hazards model was also conducted separately for patients with pre-existing diabetes and those with COVID-associated hyperglycaemia. Although many analysed factors were similarly associated with COVID-19 adverse clinical outcome in either COVID-associated hyperglycaemia or pre-existing diabetes, some qualitative or quantitative differences have emerged (Figure 1B). FBG and glucose variability were associated with adverse clinical outcome, but statistical significance was only achieved in pre-existing diabetes. A protective effect of ACE blocking was evident in pre-existing diabetes but not in COVID-associated hyperglycaemia. Among the inflammatory markers and indicators of multi-organ injury, neutrophils, haemoglobin, AST, and bilirubin were found to have more quantitative impact in subjects with COVID-associated hyperglycaemia than in those with pre-existing diabetes. Conversely, lymphocytes, creatinine, albumin, troponin T, Pro-BNP, procalcitonin, ferritin, and IL-6 were found to have more impact in subjects with pre-existing diabetes than in those with COVID-associated hyperglycaemia.

### 3.5. Post Discharge Follow-Up: Time to NT Swab Negativization and Persistence of Hyperglycaemia

As prolonged viral shedding is a risk factor for poor COVID-19 outcome and may interfere with the immunological mechanism of virus elimination, in our study population, we evaluated the time to NT swab negativization. Sex-and age-adjusted Cox proportional hazards model (HR 0.661, 95% CI 0.43–1.02; *p* = 0.063) and the Kaplan–Meier estimator log-rank test (45 days, 95% CI 37–52 vs. 35 days, 95% CI 29–41; *p* = 0.023) documented a negative association between COVID-associated hyperglycaemia and the time to NT swab negativization (Figure 3). We also evaluated fasting blood glucose during the post-discharge follow up (i.e., at the 1-, 3-, 6-, and 9-month outpatient visits). Among 121 subjects with pre-existing diabetes, 40 died, and 20 had no glucose measurements during follow up (median 6 month). Of the remaining 61 patients with a prescribed treatment for diabetes, six had normal fasting glucose (NFG) (9.8%, 90 (84.5–91.75) mg/dL), 21 had impaired fasting glucose (IFG) (34.4%, 112 (108–119) mg/dL (, and 34 had fasting glucose in the diabetes range (DFG) (55.7%, 160.5 (133–189.5) mg/dL). Among the 55 patients with COVID-associated hyperglycaemia, 23 died, and 12 had no glucose measurements during follow up (median 6 months). Of the remaining 20 patients, none had a prescribed diabetes treatment during follow up: six showed normalization of their fasting glucose (30%; 86 (84–90 mg/dL)), while 14 were confirmed as having either IFG (*n* = 8, 40%, 104 (100.2–111)) or DFG (*n* = 6, 30%; 136 (126–148) mg/dL).

## 4. Discussion

Hyperglycaemia [6] is emerging as a common feature among patients hospitalized for COVID-19 pneumonia [40]. This COVID-associated hyperglycaemia still needs to be adequately characterized since there are only few studies documenting how it differs from pre-existing diabetes in COVID-19 patients. To address this issue, we studied a cohort of 176 adult patients with COVID-19 pneumonia with a diagnosis of pre-existing diabetes or hyperglycaemia. Our study generated several interesting findings. First, patients with COVID-associated hyperglycaemia had significantly less comorbidities related to the metabolic syndrome than those with pre-existing diabetes. The main reason for this difference probably lies in the lower prevalence of obesity and a shorter hyperglycaemia duration. Consistent with this, prior to admission, only a low proportion of patients had been prescribed antihypertensive drugs, lipid-lowering agents, or antiplatelet therapy. However, whether the lower prevalence is an effect related to an underdiagnosis driven by the absence of diabetes diagnosis cannot be excluded. Similarly, the higher prevalence of neurodegenerative disease in subjects with COVID-associated hyperglycaemia could be related to the lower use of health care services and therefore lower rates of a diabetes diagnosis pre-admission in patients with a cognitive impairment. Beyond the potentially confounding factors, the lower prevalence of a metabolic signature suggests a specific pathogenesis for hyperglycaemia associated with COVID-19.

Second, patients with COVID-associated hyperglycaemia showed increased levels of inflammatory markers and indicators of multi-organ injury compared to patients with pre-existing diabetes. The correlation of hyperlgycaemia with high levels of inflammation is consistent with other conditions [41] and it is not unexpected in this context. In fact, during an overwhelming SARS-CoV-2 infection, the massive release of glucocorticoid [42] and of cytokines [43] might stimulate the gluconeogenesis and increase insulin resistance, which can contribute to the elevation in blood glucose levels. Based on this, it can be speculated that in the absence of a pre-existing diabetes, hyperglycaemia pre-existing could represent a proxy of illness severity, as also suggested by the association with and indicators of multi-organ injury.

Third, the prevalence of autoimmune biomarkers such as anti-GAD (a marker of islet autoimmunity) or interferon alpha-4 antibody (an autoimmune marker recently associated with COVID-19 severity) [44] was generally very low and did not differ between COVID-associated hyperglycaemia and pre-existing diabetes. Viral infections were described to be associated with the development of pancreatic autoantibodies leading to T1D in genetically predisposed individuals in the TEDDY study [45], and coronaviruses were identified as one of the incriminating pathogens. Our data tend to exclude that this is the case for SARS-CoV-2. The extremely low prevalence of GAD autoantibodies excludes a previously unrecognized autoimmune (pre)diabetes (i.e., LADA) as the possible cause of new onset diabetes. Moreover, the data are in agreement with our previous results on children aged 1 to 18 years participating in a public health type 1 diabetes screening program in Bavaria, where the prevalence of antibodies against SARS-CoV-2 was not associated with type 1 diabetes autoimmunity [46].

Fourth, COVID-associated hyperglycaemia was associated with a poorer clinical outcome and a delayed time to viral clearance compared to pre-existing diabetes despite the presence of superimposable humoral responses to SARS-CoV-2 and similar glucose levels. Many studies have previously reported a worse outcome in patients with new-onset hyperglycaemia without diabetes versus normoglycaemic COVID-19 patients [35,38,47], while only a few studies have reported the outcomes in patients with new-onset hyperglycaemia without diabetes versus diabetes with discordant results [33,35,38]. Of great interest is the finding of a slow viral clearance in patients with COVID-associated hyperglycaemia. Discordant data exist about differences in viral clearance or shedding in people with diabetes [48]. As increased glucose levels and glycolysis promoted Sars-CoV-2 replication in monocytes via ROS/HIFα pathway activation lead to secondary T-cell dysfunction [49], this could explain the delayed viral clearance associated with the acute hypeglycaemia in patients with COVID-associated hyperglycaemia.

Fifth, COVID-associated hyperglycaemia reversed in most patients when the viral infection resolved. As hyperglycaemia reverted in most patients when the infection resolved, it is reasonable to hypothesize that reversible transient factors, such as inflammation-induced insulin resistance, may play a role in causing hyperglycaemia in those patients [50]. Whether COVID-associated hyperglycaemia should be considered a specific clinical entity remains a matter of discussion. The use of glucocorticoids in the first wave of the pandemic was very limited and, therefore, unlikely to have played a role in the pathogenesis of hyperglycaemia. On the other hand, it is very likely that more than one cause may contribute to the hyperglycaemia associated with COVID-19. In our study, the absence of islet autoimmunity markers and the lower prevalence of comorbidities related to metabolic syndrome do suggest specific pathophysiological mechanisms responsible for hyperglycaemia. Regardless of the pathophysiological mechanism, our study documented that COVID-associated hyperglycaemia was associated with an adverse clinical outcome of SARS-CoV-2 infection and that this association was independent from the major risk factors for disease severity. As good glycemic control was indeed shown to reduce disease severity and COVID-19 mortality patients with hyperglycaemia [51], early recognition and treatment of COVID-associated hyperglycaemia may greatly benefit these patients. Therefore, our findings do strongly support the need to screen all patients with COVID-19 pneumonia for hyperglycaemia (i.e., blood glucose and/or HbA1c) at the time of admission despite a mute personal or family history of diabetes. Since blood glucose levels in our study were not different in patients with either COVID-associated hyperglycaemia or pre-existing diabetes, factors other than hyperglycaemia should be considered to explain the differences in clinical outcome. A different drug treatment prior to admission could play a role. Of note, in our study, sex-and age-adjusted Cox proportional hazards models indicated that angiotensin-converting enzyme inhibitors (ACEi) and/or angiotensin receptor blockers (ARB) treatment was significantly associated with a better clinical outcome in pre-existing diabetes. Most studies, including several meta-analyses, found no substantial differences in the risk for severe COVID-19 pneumonia associated with the prescription of common classes of antihypertensive medications [52]. On the other hand, evidence of the beneficial effects of chronic ACEi/ARB use, especially in hypertensive cohort of patients with COVID-19 pneumonia, was reported [53,54,55,56]. As the prevalence of hypertension is higher among patients with pre-existing diabetes, ACEi/ARB use may mediate a selective beneficial effect. Similarly, a role for antidiabetics could be speculated. In our study, insulin and metformin had different associations with clinical outcomes: around 60% of patients with pre-existing diabetes were taking metformin, and the treatment was associated with a better clinical outcome. Conversely, insulin treatment was associated with a worse outcome. This is in agreement with previous small studies [57,58,59,60,61,62] and was recently confirmed by a large observational nationwide study in England [63] where metformin, SGLT2 inhibitors, and sulfonylureas resulted in an association of reduced risks of the COVID-19-related mortality, whereas insulin and DPP-4 inhibitors were associated with an increased risk, and neutral results were found for GLP-1 receptor agonists and thiazolidinediones. However, because of the limitations of such real-world studies, these findings should be considered with caution and are likely due to be confounding by indication, in view of the use of different drug classes in the early and late stages of the type 2 diabetes disease trajectory (metformin is used early in the disease trajectory of type 2 diabetes, whereas insulin is typically initiated later). Few randomized clinical trials assessing the role of glucose-lowering therapies on COVID-19 outcomes in patients with type 2 diabetes are ongoing. The DARE-19 study [64] investigated the effect of the SGLT2 inhibitor dapagliflozin versus a placebo on the risk of death or organ dysfunction in patients admitted to hospital with COVID-19 and failed to demonstrate any advantages.

Our study has some limitations. First, the analysis was performed on a subcohort of 176 patients selected for having hyperglycaemia or pre-existing diabetes out of 584 subjects of our original cohort. All patients with diabetes/hyperglycaemia were included, and the age, sex, hyperglycaemia prevalence, and the clinical outcome of our cohort appears superimposable to those reported by many authors. Despite this, we cannot exclude selection bias. Second, our cohort is limited to hospitalized patients with COVID-19 pneumonia. Third, we only assessed fasting blood glucose, and we acknowledge that more specific indicators of beta cell function, such as serum insulin and or C peptide levels, should have been measured.

## 5. Conclusions

COVID-associated hyperglycaemia is emerging as a complication of Sars-CoV-2 infection and this clinical entity still needs to be adequately characterized in comparison to pre-existing diabetes. It is clear from our study that patients with COVID-associated hyperglycaemia had increased levels of inflammatory markers and indicators of organ injuries associated with a poorer clinical outcome and a delayed viral clearance compared to pre-existing diabetes. As good glycemic control was demonstrated to decrease disease severity and mortality in COVID-19 patients with hyperglycaemia [51], early recognition and treatment of COVID-associated hyperglycaemia may greatly benefit these patients. This strongly supports the need to screen all COVID-19 patients for hyperglycaemia at the time of admission despite a mute personal or family history of diabetes and to treat them in order to reach and maintain a good glycemic control during hospitalization for COVID-19 pneumonia.

## Figures and Tables

**Figure 1 biology-10-00754-f001:**
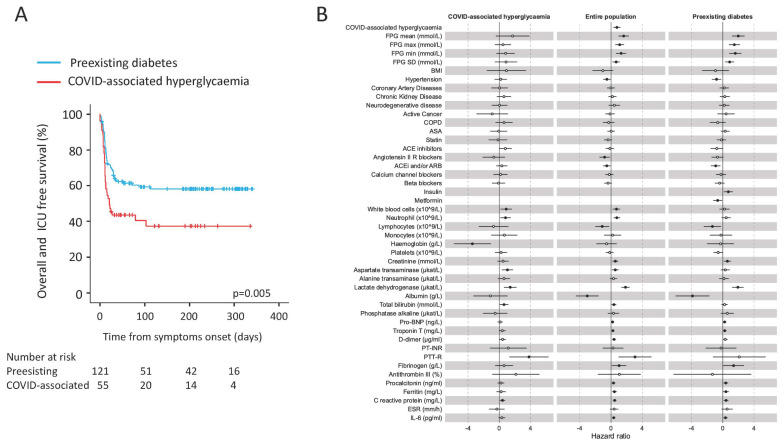
(**A**,**B**) Adverse clinical outcome in patients with Sars-CoV-2 infection with either pre-existing diabetes or COVID-associated hyperglycaemia. Kaplan–Meier estimates of survival without adverse clinical outcome for 176 patients with COVID-19 pneumonia (**A**). Survival without adverse clinical outcome (defined by composite endpoint of admission to ICU or death, whichever occurred first) was estimated for patients with COVID-associated hyperglycaemia (*n* = 55) or pre-existing diabetes (*n* = 121). The log-rank test was used to test differences between the two groups. Crosses indicate censored patients (censoring for death or end of follow-up). The forest plots (**B**) show the hazard ratios for survival without adverse clinical outcome according to presence of new-onset or pre-existing diabetes. Univariate Cox regression analysis adjusted for sex and age. Dots represent the HR, lines represent 95% confidence interval (CI), and solid dots indicate *p* < 0.05.

**Figure 2 biology-10-00754-f002:**
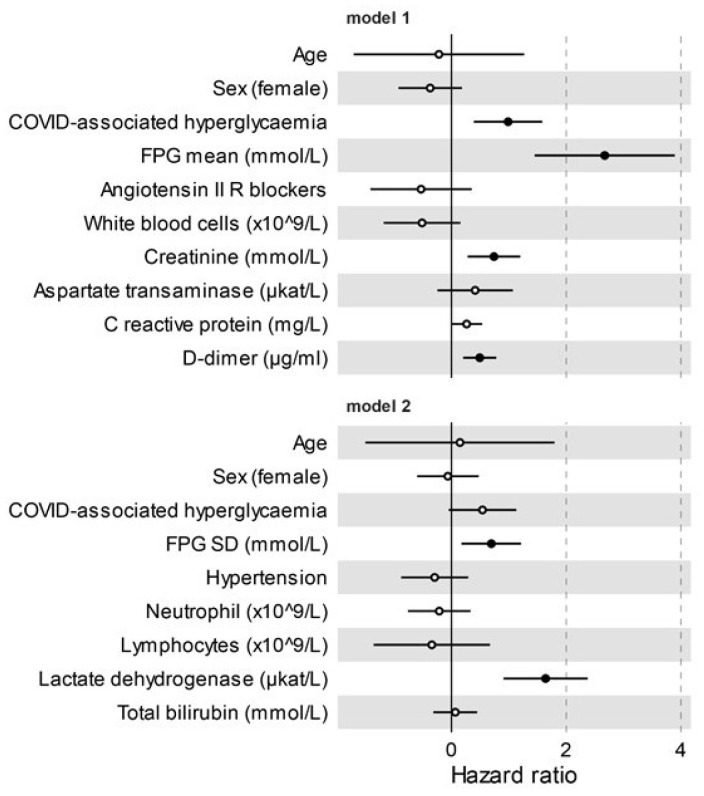
Adverse clinical outcome in patients with Sars-CoV-2 infection. Multivariate Cox regression analysis adjusted for sex and age, including variables significant at the level of <0.1 in the univariate analysis. Variables were excluded from multivariate Cox regression if they showed substantial biological redundancy with other variables (e.g., aspartate aminotransferase vs. alanine aminotransferase) or had data obtained for fewer than 78% of patients. All of the tested variables are reported. Dots represent the HR, lines represent 95% confidence interval (CI), and solid dots indicate *p* < 0.05.

**Figure 3 biology-10-00754-f003:**
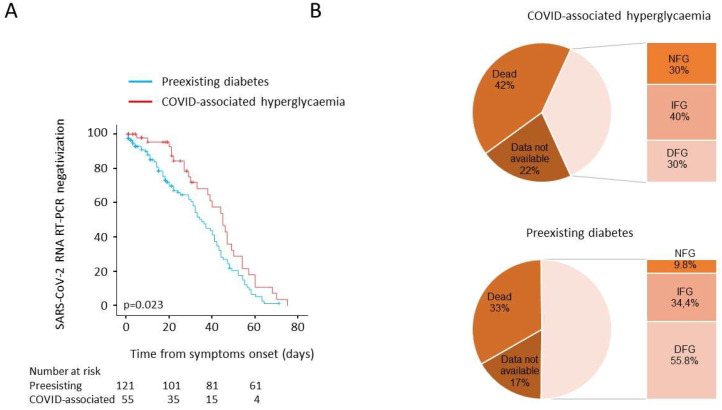
(**A**,**B**) Post discharge follow-up: time to NT swab negativization and persistence of hyperglycaemia. Kaplan–Meier estimates of the time to NT swab negativization for 176 patients with COVID-19 pneumonia (**A**). NT swab negativization was estimated in patients with COVID-associated hyperglycaemia (*n* = 55) or pre-existing diabetes (*n* = 121). The log-rank test was used to test differences between groups. Crosses indicate censored patients (censoring for death or end of follow-up). Fasting blood glucose during the post discharge follow-up (**B**). Valid glucose measurements during follow up (median 6 month) was available for 61 out of 121 and 20 out of 55 patients with pre-existing diabetes or COVID-associated hyperglycaemia, respectively. NFG: normal fasting glucose, <5.6 mmol/L; IFG: impaired fasting glucose 5.6-6.9 mmol/L; DFG: diabetes fasting glucose ≥7 mmol/L.

**Table 1 biology-10-00754-t001:** Baseline characteristics of study cohort according to either pre-existing diabetes or COVID-associated hyperglycaemia.

	Pre-Existing Diabetes	COVID-Associated Hyperglycaemia	*p*	Missing
N	121	55		
Age, years	71 (62–77.5)	69 (54–78)	0.324	0
Sex, male [N (%)]	84 (71.2)	37 (63.8)	0.387	0
BMI	29.16 (26.12–33.8)	26.52 (23.87–30.12)	0.014	20
- <25 [N (%)]	23 (20.5)	14 (31.8)	0.036	
- 25–30 [N (%)]	36 (32.1)	19 (43.2)		
- >30 [N (%)]	53 (47.3)	11 (25)		
Ethnicity [N (%)]			0.998	0
- Caucasian	106 (87.6)	48 (87.3)		
- Hispanic	7 (5.8)	3 (5.5)		
- Asian	4 (3.3)	2 (3.6)		
- African	4 (3.3)	2 (3.6)		
Comorbidities [N (%)]				0
- Hypertension	91 (75.2)	32 (58.2)	0.033	
- Coronary Artery Diseases	35 (28.9)	6 (10.9)	0.012	
- COPD	16 (13.2)	5 (9.1)	0.617	
- Chronic Kidney Disease	32 (26.4)	7 (12.7)	0.05	
- Cancer	23 (19)	3 (5.5)	0.021	
- Neurodegenerative disease	7 (5.8)	8 (14.5)	0.078	
Preadmission treatment [N (%)]				0
- Metformin	71 (58.7)	0		
- Sulfonylureas	12 (9.9)	0		
- Repaglinide	3 (2.5)	0		
- Acarbose	3 (2.5)	0		
- Thiazolidinediones	1 (0.8)	0		
- DPP-4 inhibitors	14 (11.6)	0		
- GLP-1 agonist	3 (2.5)	0		
- SGLT2 inhibitors	4 (3.3)	0		
- Insulin	41 (33.9)	0		
- ASA	47 (38.8)	7 (12.7)	<0.001	
- Statin	44 (36.4)	6 (10.9)	0.001	
- ACE inhibitors	28 (23.1)	11 (20)	0.699	
- Angiotensin II R blockers	28 (23.1)	6 (10.9)	0.065	
- ACEI and/or ARB	54 (44.6)	17 (30.9)	0.099	
- Calcium channel blockers	35 (28.9)	9 (16.4)	0.091	
- Beta blockers	43 (35.5)	13 (23.6)	0.162	
Admission to the hospital				0
Median time from symptoms to admission, days	6 (2–9.5)	7 (3–10)	0.484	
Symptoms at onset [N (%)]				
- Fever	93 (78.2)	46 (85.2)	0.31	
- Dyspnea	79 (66.4)	37 (68.5)	0.86	
- Cough	46 (38.7)	21 (38.9)	0.99	
- fatigue/malaise	35 (29.4)	10 (18.5)	0.14	
- hypo/dysgeusia	19 (17.3)	8 (14.8)	0.82	
- hypo/anosmia	17 (14.3)	8 (14.8)	0.99	
- myalgia/arthralgia	19 (16)	7 (13)	0.82	
- headache	12 (10.1)	1 (1.9)	0.066	
- chest pain	12 (10.1)	2 (3.7)	0.23	
- diarrhea	16 (13.4)	6 (11.1)	0.81	
- sore throat	5 (4.2)	6 (11.1)	0.10	
- vomiting/nausea	9 (7.6)	6 (11.1)	0.56	
- conjunctivitis	8 (6.7)	5 (9.3)	0.55	
- abdominal pain	5 (4.2)	2 (3.7)	0.99	
- skin rash	2 (1.9)	0	0.99	

**Table 2 biology-10-00754-t002:** Clinical laboratory profile at hospital admission and clinical outcome.

	Pre-Existing Diabetes	COVID-Associated Hyperglycaemia		Missing
N	121	55	*p*	
Clinical outcomes				0
Median follow up, days (95%CI)	222 (202–242)	190 (98–282)	0.065	
After admission [N (%)]				
- Discharged	79 (65.3)	32 (58.2)		
o Not hospitalized	7 (8.9)	4 (12.5)	0.008	
o Hospitalized ≤ 7 days	14 (17.7)	1 (3.1)		
o Hospitalized > 7 days	51 (64.6)	17 (53.1)		
o ICU	7 (8.9)	10 (31.3)		
- Dead	42 (34.7)	23 (41.8)		
o After ICU	11 (26.2)	11 (47.8)	0.103	
Median hospital stay, days	15.5 (7–29)	20 (13–31)	0.098	
Adverse clinical outcome [N (%)]	49 (40.5)	33 (60)	0.022	
Swab negativization, days (95%CI)				
Median time from symptoms,	35 (29–41)	45 (37.5–52.5)	0.023	0
Fasting glucose (mmol/L)				
- Median	8.38 (6.55–10.99)	7.97 (7.43–9.19)	0.882	0
- Max	11.04 (8.21–14.62)	9.49 (7.93–11.59)	0.109	
- Min	6.1 (4.85–8.1)	7.04 (5.38–7.71)	0.238	
- Glucose variability (SD)	2.44 (1.3–3.64)	1.78 (1.01–3)	0.158	
- N° of determinations	3 (2–7)	2 (1–7)	0.043	
Glycated haemoglobin	49.5 (42.25–57.75)	37 (36–47)	0.005	111
Laboratory at admission *:				
White blood cells (×10^9^/L)	7.1 (5.4–10.4)	9.7 (6.1–14.3)	0.004	0
- Neutrophil (×10^9^/L)	5.2 (3.9–8.1)	8.4 (4.85–12.9)	0.003	2
- Lymphocytes (×10^9^/L)	0.9 (0.6–1.3)	0.9 (0.6–1.35)	0.55	2
- Monocytes (×10^9^/L)	0.5 (0.3–0.7)	0.5 (0.3–0.7)	0.5	0
- Haemoglobin (g/L)	12.5 (10.75–13.5)	12.9 (11.5–14.6)	0.042	0
- Platelets (×10^9^/L)	227 (166–298)	228 (160–344)	0.62	0
Creatinine (µmol/L)	95.5 (74.3–134.8)	93.7 (76.9–137.9)	0.99	0
Aspartate transaminase (µkat/L)	0.63 (0.42–1.14)	0.95 (0.65–1.39)	0.001	0
Alanine transaminase (µkat/L)	0.53 (0.3–0.93)	0.78 (0.47–1.24)	0.012	0
Lactate dehydrogenase (µkat/L)	5.92 (4.42–8.17)	8.04 (5.35–16.63)	0.001	0
Albumin (g/L)	27.9 (24.48–30.98)	26.5 (22.7–29.7)	0.17	57
Total bilirubin (µmol/L)	9.23 (5.39–14.87)	11.62 (8.16–18.81)	0.054	5
Phosphatase alkaline (µkat/L)	1.27 (0.97–1.92)	1.26 (0.95–1.81)	0.65	29
Pro-BNP (ng/L)	529 (288–1663)	682 (182–2429)	0.72	64
Troponin T (µg/L)	19 (11.42–45.57)	26.6 (11.02–88.67)	0.29	44
D-dimer (µg/mL)	1.69 (0.68–3.83)	2.88 (1.04–6.62)	0.18	39
PT-INR	1.12 (1.04–1.2350)	1.19 (1.06–1.29)	0.11	15
PTT-R	1 (0.92–1.1)	0.98 (0.91–1.01)	0.83	15
Fibrinogen (g/L)	607 (483–746)	570 (433–741)	0.70	85
Antithrombin III (%)	91 (83.7–100.2)	89.5 (76.2–110)	-	142
Procalcitonin (ng/mL)	0.61 (0.31–1.38)	0.95 (0.49–3.58)	0.014	42
Ferritin (µg/L)	823 (462–1499)	1254 (585–2433)	0.063	40
C reactive protein (mg/L)	93.5 (28.35–172.25)	113.6 (31.1–204.9)	0.39	0
ESR (mm/h)	73 (39–106)	68 (48–87)	-	98
IL-6 (pg/mL)	56.4 (23.1–173)	59.8 (27.12–163.25)	0.96	74
Humoral immune response [N (%)]				
Sampling time from symptoms, days	9 (4.75–13)	10 (7–17.75)	0.20	27
- Anti-GAD antibody	3 (3)	0 (0)	0.55	
- Interferon alpha-4 Antibody	4 (4)	4 (8.3)	0.27	
- SARS-Cov2 RBD IgG	53 (52.5)	27 (56.3)	0.73	
- SARS-Cov2 RBD IgM	56 (55.4)	29 (60.4)	0.60	
- SARS-Cov2 RBD IgA	57 (56.4)	34 (70.8)	0.11	
- SARS-Cov2 S1 + S2 IgG	60 (59.4)	31 (64.6)	0.59	
- SARS-Cov2 S1 + S2 IgM	79 (78.2)	35 (72.9)	0.54	
- SARS-Cov2 S1 + S2 IgA	76 (75.2)	42 (87.5)	0.13	
- SARS-Cov2 NP IgG	66 (65.3)	35 (72.9)	0.45	

* *p*-values were not reported for variables that had >50% of missing data.

## Data Availability

The datasets generated during and/or analyzed during the current study are not publicly available due to concerns about data confidentiality but are available from the corresponding author upon reasonable request.

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
