# Peer review of "Pre-Existing Diabetes and COVID-Associated Hyperglycaemia in Patients with COVID-19 Pneumonia"

_biology, 2021, doi:10.3390/biology10080754_

Round 1
Reviewer 1 Report
The manuscript entitled " Preexisting Diabetes and COVID-Associated Hyperglycaemia 2
in Patients With COVID-19 Pneumonia". Title, abstract and overall rationale of work to some extent is good and interesting. However, there are some minor concerns, which needs to be addressed and needs minor revision.
1) Introduction section should be elaborate.
2) Figure 1 B need to increase resolution.
3) I would suggest the authors to enhance your theoretical discussion and arrives your debate or argument.
4) Line number 275-283 should be non-italic and also correct the font size.
5) Author must elaborate the conclusion section. This section should present at least in one 250-300 words paragraph.
Reviewer 2 Report
Recent studies have reported that COVID-associated hyperglycaemia is associated with a poorer outcome compared to that of the normoglycemic individuals.
However, whether COVID-associated hyperglycaemia is associated with a poorer clinical outcome compared to preexisting diabetes is still an open question, with conflicting findings. Also, data on islet autoimmunity prevalence, anti SARS-CoV-2 antibody responses, and timing for viral clearance are still missing.
To address this gap in our knowledge, the authors studied a cohort of 176 adult patients with confirmed COVID-19 pneumonia, with a diagnosis of preexisting diabetes or hyperglycaemia.
No differences between preexisting diabetes and COVID-associated hyperglycaemia were evident for symptoms at admission, humoral response against SARS-CoV-2 or autoantibodies to glutamic acid decarboxylase or interferon alpha-4.
However, they found that COVID-associated hyperglycaemia was independently associated with the risk of adverse clinical outcome defined as ICU admission or death (HR 2.11, 95% CI 1.34-3.31; p=0.001), even after adjustment for age, sex and other selected variables associated with COVID-19 severity.
Furthermore, the authors documented a negative association (HR 0.661, 95% CI 0.43-1.02; p=0.063) between COVID-associated hyperglycaemia and the time to swab negativization.
From the data the concluded that recognition of hyperglycaemia as a specific clinical entity associated with COVID-19 pneumonia is relevant for early and appropriate patient management and close monitoring for the progression of disease severity.
This study used a well-characterized population pool from two clinical trials to reveal that although the prevalence of autoimmune biomarkers as anti-GAD (a marker of islet autoimmunity) or interferon alpha-4 antibody (an autoimmune marker recently associated with COVID-19 severity) was generally very low and did not differ between COVID-associated hyperglycaemia and preexisting diabetes
- Patients with COVID-associated hyperglycaemia had significantly less comorbidities related to the metabolic syndrome than those with preexisting diabetes. Patients with COVID-associated hyperglycemia also had lower prevalence of obesity and a shorter duration of hyperglycemia.
- Patients with COVID-associated hyperglycaemia had a higher prevalence of neurodegenerative disease and suggested that it could be related to lower utilization of health care and therefore lower rates of diagnosis of diabetes pre admission.
- They also found that patients with COVID-associated hyperglycaemia showed increased levels of inflammatory markers and indicators of multi-organ injury compared to patients with preexisting diabetes.
- COVID-associated hyperglycaemia was associated with a poorer clinical outcome and a delayed time to viral clearance compared to preexisting diabetes, despite the presence of a super imposable humoral responses to SARS-CoV-2 and similar glucose levels.
- COVID-292 associated hyperglycaemia reversed in most patients when the viral infection resolved.
However, the underlying cause for the COVID-associated hyperglycaemia was not resolved in this manuscript since the use of glucocorticoids in the first wave of the pandemic was very limited and, therefore, unlikely to have played a role in the pathogenesis of hyperglycaemia. They did suggest that inflammation-induced insulin resistance, may play a role in causing hyperglycemia in those patients.
Overall, this study documented that COVID-associated hyperglycaemia was associated with an adverse clinical outcome of SARS-CoV-2 infection and that this association was independent from the major risk factors for disease severity. Addition explanations other than lower rates of diagnosis of diabetes pre admission to help explain the higher prevalence of neurodegenerative disease.
This study will be of significant interest to readers
Reviewer 3 Report
The present manuscript compared the clinical presentations and major outcomes of hospitalized COVID-19 patients with preexisting diabetes (n=112) and COVID-associated hyperglycaemia (n=55), and highlighted the association between COVID-associated hyperglycaemia and worse progression. In addition, the manuscript presented data regarding the humoral response to SARS-Cov2 and timing for viral clearance among the two groups. The results are generally well presented, however, several issues need to be addressed.
- The authors mentioned patients with COVID-associated hyperglycaemia showed increased levels of inflammatory markers and organ injury indicators comparing to patients with preexisting diabetes, is it also possible that their worse outcome and slower viral clearance are both effects related to more severe COVID-19 illness at the time of hospital admission? The reviewer also wonders whether COVID-19 of higher severity is more likely to cause hyperglycaemia.
- The data regarding antidiabetics in Table 1 and Figure 1 seems to suggest the usage of two most common antidiabetics, insulin and metformin, had different association with clinical outcomes. To deliver further insights on glycemic control for patients with diabetes or likely to develop diabetes, the authors should include a brief discussion on glucose-lowering treatment based on their data and some previous publications (for examples, PMID: 33188364, 32409498, 32472191, 33248471).
- Page 8, line 212, please specify the variants applied in the two different multivariate Cox regression models, either in the text or in figure legends.
- Page 8, line 218, the differences in the risk factors for adverse clinical outcome in two groups are reported using non-descriptive wording “some qualitative or quantitative differences emerged”, which is not informative enough.
- Page4, line 140, section 3.1, “Among the 176 cases included, ... 11 (6.3%) were discharged ... 137 (77.8%) were treated with non-invasive ventilation and 39 (22.2%) were admitted to the ICU...”, since 11+137+39 equals to 177 but not 176, this may cause confusion to the readers. Presumably this is an error?
Round 2
Reviewer 3 Report
The authors have addressed the comments